# Surface Electromyography Combined with Artificial Intelligence in Predicting Neuromuscular Falls in the Elderly: A Narrative Review of Present Applications and Future Perspectives

**DOI:** 10.3390/healthcare13101204

**Published:** 2025-05-21

**Authors:** Yuandan Liao, Gang Tan, Hui Zhang

**Affiliations:** 1Department of Orthopaedic Surgery, West China Hospital, Sichuan University, No. 37, Guoxue Avenue, Chengdu 610041, China; liaoyuandan101@163.com (Y.L.); tg-828@126.com (G.T.); 2Department of Orthopedics, West China School of Public Health and West China Fourth Hospital, Sichuan University, Chengdu 610041, China

**Keywords:** sEMG, artificial intelligence, neuromuscular falls, elderly

## Abstract

**Background**: Falls among the elderly are a major public health concern, leading to increased disability and mortality. Traditional protective measures are important, but early detection and prevention are equally critical. Surface electromyography (sEMG) signals, which assess muscle electrical activity, can indicate a fall risk by detecting muscle weakness or instability. **Objective**: This narrative review synthesizes the research progress of sEMG in predicting neuromuscular falls among the elderly. Our goal is to explore the innovative application and development potential of the integration of sEMG and artificial intelligence (AI) in fall prevention among the elderly. **Methods**: A systematic search of PubMed, IEEE Xplore, and Web of Science (2013–2023) was conducted using the following keywords: artificial intelligence, wearable, sEMG, neuromuscular, and fall prediction. The inclusion criteria prioritized studies integrating sEMG with AI for elderly fall risk assessments, while non-empirical or non-English studies were excluded. **Results**: AI algorithms hold significant potential in medical applications, and studies on predicting neuromuscular falls in the elderly using sEMG signals have made notable progress. However, limitations include a reliance on simulated data, a lack of standardized models, sensor inaccuracies, and a focus on prediction rather than prevention. To address these challenges, this study proposes collecting authentic sEMG signals from elderly individuals with fall histories and healthy controls. By leveraging AI to develop predictive models and designing a portable sEMG acquisition and analysis system tailored for elderly communities, real-time fall risk predictions and early warnings can be achieved, thereby reducing fall incidences among the elderly. **Conclusions**: The combination of sEMG and AI presents a substantial promise for predicting neuromuscular falls in the elderly. Future research should prioritize validating models in real-world settings, refining sensor technology and signal processing techniques, and shifting focus toward comprehensive preventive strategies rather than mere prediction. These advancements could significantly enhance the quality of life and health outcomes of the elderly, while alleviating burdens on families and healthcare systems.

## 1. Introduction

In recent years, due to the extension of life expectancy and the decline of the birth rate, the size and proportion of the elderly population has been increasing, and by 2050, the global population aged 60 and above is expected to reach two billion. According to WHO statistics, China’s aging rate is much faster than other countries, and by 2050, China may become the country with the largest aging population in the world [1]. However, with age, human body functions gradually decline and muscle strength weakens, which often leads to falls. WHO statistics show that 28–35% of people aged 65 and above have falls every year, the data are 32–42% in the 70 years old and above category [2,3]. The risk of falls in older adults is influenced by multifactorial determinants, encompassing both internal and external factors that collectively contribute to an increased likelihood of falls. Environmental hazards, such as poor lighting, uneven surfaces, and slippery floors, are significant contributors [4]. Evidence from epidemiological studies indicates that key internal risk factors significantly associated with fall incidence include gender disparities, chronic comorbidities, functional disability, compromised activities of daily living (ADL), and diminished physical function (PF) [5]. In particular, sarcopenia and fibromyalgia are highly prevalent among the aging population. These conditions can lead to impaired muscle activation, reduced muscle strength, and increased fatigue, severely compromising postural control and mobility and thus markedly elevating the risk of falls [6,7]. When the elderly fall, there is a high risk of events such as hip bone fractures, vertebral compression fractures, and so on, which further aggravates the disability status of patients, endangers the lives of the elderly, and brings huge economic burdens to society and families [8,9].

Electromyography (EMG) is a technique which measures the electrical activity generated by muscles, while surface electromyography (sEMG) refers to the electromyography collected on the surface of the skin [10,11]. When the human body is exercising, the motor neuron sends out a signal, and the signal will generate a bioelectric current during the transmission of the human body. The sEMG signal is the sum of these electrical signals, which contains a lot of information associated with limb movement. The sEMG signals produced by the same muscle are different when performing different actions. When the same action is repeated, the sEMG signal waveform changes produced by the muscles are very similar, even if there are certain differences between different individuals, there are still rules to follow [12]. Based on this, to a certain extent, sEMG can be used as the basis for identifying limb movements and reflecting muscle movement intentions [13]. sEMG, as an advanced non-invasive technique for biological signal acquisition, has been widely applied in clinical medicine, rehabilitation, sports science, and ergonomics. In clinical research, sEMG is used for neuromuscular assessment and disease diagnosis [14], while in rehabilitation, it assists in monitoring and optimizing training [15]. In sports science, sEMG supports the analysis of muscle activation, performance enhancement, and injury prevention [16]. In ergonomics, it aids in evaluating muscle load and fatigue to inform safer workplace designs [17]. In the prediction of falls among the elderly, sEMG can be utilized to analyze muscle activation patterns, variations in muscle strength, and levels of fatigue, all of which are critical factors in maintaining postural stability and are closely associated with fall risk. Compared with traditional scale-based assessments, which rely on subjective evaluations at a single time point, sEMG offers a more sensitive, objective, and dynamic evaluation of neuromuscular function in older adults.

Artificial intelligence (AI) has demonstrated broad prospects for development across multiple fields. In clinical medicine, the integration of surface electromyography (sEMG) with AI has achieved significant advances, with wide applications in neuromuscular function assessment, motor control analysis, disease diagnosis, and rehabilitation interventions. For example, Moore et al. [18] developed a wearable sEMG system capable of predicting gait freezing episodes in patients with Parkinson’s disease, thereby enabling early intervention to reduce injury risks. Shefa et al. [19] utilized a smart ankle–foot orthosis integrating sEMG and inertial sensors, combined with the Internet of Things (IoT) and machine learning technologies, to achieve gait pattern recognition and accurately predict improvements in gait. Guo et al. [20] designed a wearable rehabilitation glove incorporating four sEMG sensors and deep learning algorithms, which effectively assists patients with mild or partial paralysis in the recovery of hand motor functions.

While previous studies have explored the use of sEMG or AI independently in clinical and rehabilitation contexts, few have specifically addressed their combined application for the early prediction of neuromuscular falls in the elderly. Therefore, the purpose of this review is to synthesize the current research progress on the integration of sEMG and AI for fall prediction. By emphasizing proactive early warning rather than reactive detection, we aim to highlight an underexplored yet promising direction in fall prevention. This review further discusses the potential for developing portable, wearable systems that leverage this integration to serve elderly communities in real-world settings.

## 2. Methods

We conducted a literature search on PubMed, Medline, Web of Science, IEEE Xplore, and Google Scholar. A search string consisting of the following keyword combinations was adopted: “elderly” AND (“artificial intelligence” OR “AI” OR “machine learning” OR “deep learning” OR “neural networks”) AND (“sEMG” OR “surface electromyography”) AND “fall”. The time range of the publications considered was 12 years, from 2013 to 2025. Studies were included if they (i) involved elderly individuals with a focus on fall risk; (ii) applied sEMG in combination with artificial intelligence to analyze neuromuscular signals for fall prediction or early warning; and (iii) were published in English. Excluded studies were those that (i) did not involve elderly individuals or focused on other age groups; (ii) did not utilize sEMG for signal analysis, such as those relying solely on alternative sensors or technologies; (iii) did not incorporate artificial intelligence or machine learning techniques in the analysis of neuromuscular signals; (iv) were not accessible as a full text; or (v) were book chapters, opinion pieces, commentaries, or other non-data-driven literature.

A total of 2670 articles were initially identified through database searching. After removing 763 duplicates and 1670 clearly irrelevant articles based on titles and abstracts, 237 articles remained for full-text retrieval. A total of 2 articles were unavailable, leaving 235 articles for full-text assessment. Following a detailed evaluation against the eligibility criteria, 42 articles were finally included in this review. The detailed process of article selection is shown in Figure 1.

## 3. Research Status of Global Elderly Falls

As mentioned in the introductory section, falls occur at a high rate among people aged 65 and over, with more than a quarter of older adults falling each year, mainly due to the decline in the health status of older adults [21,22]. For example, muscle atrophy and decreased muscle strength in the elderly lead to the decreased walking ability of patients. In addition, as age increases, the mass and strength of human muscles are constantly decreasing [23]. According to statistics, the prevalence of sarcopenia in the elderly aged 60–70 is 5–13%, and with age, the prevalence of sarcopenia will also increase [24]. Yeung et al. [25] showed that muscle weakness (mainly lower limb muscle weakness) is positively correlated to the inability to maintain standing balance, which is an important risk factor for falls. And many elderly suffer from diabetes, high blood pressure, Alzheimer’s disease, and other chronic illnesses, as well as long-term “living with illness” and long-term use of drugs. Studies have shown that opioids, benzodiazepines, and certain antipsychotic drugs will increase the risk of falls [26]. According to incomplete statistics, one-third of the elderly fall at least once a year in the community, and nearly half of them will fall repeatedly [27,28]. The incidence and severity of associated injuries are increasing, and it has now become the third leading cause of disability in older adults [29,30]. Research by Bergen et al. showed that in 2014, 28.7% of the elderly in the United States had a fall event, and 37.5% of them were limited in activities or required medical care due to the fall [31,32]. In addition, the study also showed that women are more likely to fall and are more likely to be injured than men [31]. According to the research of Moreland et al. [33], in terms of emergency visits related to falls, the visit rate of women is higher than that of men, and the rate of visits in rural areas is higher than that in cities.

Injuries from falls include abrasions, bruises, and more serious injuries such as head injuries and hip fractures [34], with hip fractures being the most serious and reducing independence in older adults [35], even called “the last fracture in life”. In the study of osteoporosis, it was shown that hip fractures are caused by age-related bone loss or osteoporosis [36,37]. Studies have shown that nearly one in five falls result in serious injury, which is closely related to the high mortality and morbidity of the elderly, and falls are also one of the causes of accidental death in the elderly [38]. In particular, falls are most common among people aged 70 years and older and are still rising sharply [39]. There are also some elderly people who miss the best medical rescue time because after falling they are not detected in time, lying on the ground for a long time or are even comatose, and it even increases the risk of pneumonia, pressure ulcers, and death. For frail older adults, falls can have health consequences, whether they are injured or not. Even if there is no substantial physical damage at the time, it will cause a serious psychological burden on the elderly and increase their sense of fear. Because the elderly are afraid of falling again, they will avoid going out and physical activities, resulting in a decline in physical function, thus falling into a vicious circle [40]. In addition, studies have shown that older adults who have had a fall event are more likely to have another fall [41]. In addition to physical injury and psychological burden, the occurrence of falls will also cause serious economic burdens. According to statistics, more than 800,000 patients are hospitalized due to falls in the United States every year, and 37.3 million falls require medical care each year [42]. In 2015, the total medical costs of falls totaled more than USD 50 billion, with Medicare and Medicaid covering 75% of these costs [43].

Previous studies have shown that the consequences of a fall are closely related to the type and severity of the fall, including the height, direction, the mechanism of the fall, the impact force generated by the impact, and the anatomical location of the impact [44]. Therefore, if the occurrence of falls of the elderly can be predicted in advance, thereby reducing the incidence of falls or using devices to buffer the impact generated by the fall and protect the parts that may be injured, this will greatly reduce the economic burden and social burden caused by the falls of the elderly. At present, most of the wearable fall warning systems apply the principle of inertia, among which the accelerometer is the main one [45]. When the user’s movement range is too large in daily life activities and the acceleration changes, or when the user is in an elevator and other scenarios, false alarms will occur. Of course, any system will have the possibility of false positives. In the following, we also use a certain amount of space to mention how to minimize the occurrence of false positives. In addition, whether wearable devices can exert their maximum effectiveness also depends on the ability of guardians to supervise their active use by the elderly group and respond in time when they receive an alert. Or when the device is connected to a service agency or community [46], service personnel can take each alarm seriously and not slack off, even if there are many false alarms during the period [47], so that the entire monitoring system can truly achieve the purpose of caring for the elderly and the fall warning system is effective. However, as far as the fall warning system itself is concerned, it must have a positive effect. For example, Kosse et al. [48] found that the evidence of whether the sensor system can effectively prevent the elderly from falling and prevent related injuries are not consistent after sorting out twelve related studies. But while the sensors themselves may not reduce the incidence of falls, they may reduce serious injuries from falls.

## 4. The Application of Artificial Intelligence and sEMG in the Medical Field

### 4.1. The Application of Artificial Intelligence in the Medical Field

AI refers to the technology that enables computers and machines to emulate human learning, understanding, decision-making, and creativity. AI has evolved from early systems based on symbolic logic reasoning to techniques that emulate human learning through machine learning. With advancements in computational power and the expansion of datasets, deep learning technologies have further automated the processes of feature extraction and decision-making [49]. With the advancement of science and technology, AI and machine learning continue to penetrate into the medical field. From disease diagnosis and treatment to rehabilitation, and even medical record storage, artificial intelligence continues to change and improve the level of automation to simplify medical procedures to provide better medical services for patients. In recent years, AI has become a research hotspot in the field of healthcare.

AI plays a critical role in genetic diagnosis, particularly in accelerating the diagnostic process and improving accuracy. For example, the Eye2Gene algorithm uses deep convolutional neural networks to predict disease-causing genes for inherited retinal diseases from retinal images, aiming to accelerate and broaden the access to diagnoses [50]. AI can also be combined with medical imaging to help with the early diagnosis of diseases. A multi-task explainable deep-learning model, ExPN-Net, was proposed for pulmonary nodule diagnosis. This model not only predicts lesion malignancy but also identifies and visualizes nodule characteristics, improving both the diagnostic accuracy and interpretability, which supports radiologists in clinical decision-making [51]. In addition, AI is also an auxiliary diagnostic method for Parkinson’s, brain tumors, lung cancer, diabetes, and some epidemic diseases. AI has emerged as an exceptional tool in personalized medicine, significantly enhancing the precision of diagnostic and therapeutic decision-making. Phenotypic Personalized Medicine (PPM), utilizing the Quadratic Phenotypic Optimization Platform (QPOP) and CURATE.AI, leverages individual-specific data for optimizing drug combinations and dosing strategies, offering a more precise alternative [52]. AI has also penetrated into various medical fields such as drug research and development [53], health management [54], hierarchical diagnosis [55], and so on.

The integration of AI with sEMG demonstrates a substantial application potential. AI facilitates the automated processing and analysis of large-scale sEMG datasets, thereby enhancing the efficiency and accuracy of the signal interpretation. By leveraging machine learning algorithms, AI can identify patterns and extract key features from sEMG signals, enabling precise assessments of neuromuscular activity [56]. Beyond diagnostic applications, this approach can also be extended to rehabilitation monitoring and the development of personalized intervention strategies, thereby providing more targeted healthcare solutions for the elderly population.

### 4.2. Research on sEMG

sEMG is a type of human bioelectrical signal. The electromyographic signal occurs after the brain’s intention is generated, but before the actual muscle contraction [57]. sEMG provides information related to muscle activity [58] and can also provide the state of the nervous system that serves the muscles [59]. Because in each specific daily life activity, such as sitting, standing, and walking, the muscle functions and sEMG signals produced by lower limb movements are different [60]. Moreover, the generation of sEMG is 30~150 ms earlier than the occurrence of limb movements, so it is possible to predict an upcoming limb action based on sEMG [61]. By obtaining, classifying and identifying electromyogramsto predict the intention of human limbs, sEMG can be applied to various fields such as prosthetic movement control, limb rehabilitation evaluation, gait analysis, neuromuscular function evaluation, and it has gradually become an important trend for the future [62,63].

Gait analysis based on sEMG is an important diagnostic and rehabilitation method for musculoskeletal diseases [63]. Wei et al. [59] recruited seven healthy volunteers and performed 15 experiments at three different speeds, and a gait analysis was carried out by collecting the sEMG signals and EEG of eight muscles. The results show that sEMG achieves high accuracy in identifying seven gait stages, and the gait stage is easier to identify when the speed is faster, but the study only considers three walking speeds. In the future, we are looking forward to more gait phase recognition studies at different speeds.

Beyond movement analysis, wearable sEMG devices have also been explored for lower limb rehabilitation in the elderly. Chen et al. [64] designed a wearable sEMG device for the lower limb rehabilitation of the elderly at home. The device is divided into two parts, and the upper part is used to monitor the EMG signal of the quadriceps muscle; the lower part is used to monitor the EMG signal of the anterior tibia and gastrocnemius muscle. The main function is to help the elderly to train the lower limbs, and it can also collect sEMG signals for medical diagnosis. The study recruited three volunteers who had never used wearable devices for testing to initially understand the possible problems during the use of the device, and then recruited eight elderly people with an average age of 70.5 years in the community to perform lifting, tilting, and hooking three limb rehabilitation actions [65]. And this study understands the elderly’s subjective thoughts, willingness to use the device, and other problems encountered during use.

Recent studies have explored the use of wearable sEMG devices for the real-time monitoring of the user status, enabling adjustments to rehabilitation training or providing preventive alerts. For example, Nouredanesh et al. [66] developed a wearable sEMG-based system capable of detecting compensatory balance reactions (CBRs) during walking. Their system demonstrated an excellent accuracy (92.35%) in distinguishing CBRs from normal walking patterns and a good accuracy (84.60%) in classifying multiple gait patterns. This research highlights the feasibility of combining sEMG signals with wearable sensors for continuous gait and balance monitoring, paving the way for personalized fall risk assessments and interventions.

sEMG can be used for continuous monitoring, adaptive training, and dynamic feedback, providing a more proactive strategy for fall prevention. Integrating sEMG technology into predictive, rehabilitation, and feedback systems has the potential to enhance the mobility, safety, and quality of life of elderly individuals.

## 5. The Insufficiency of Existing Research on Neuromuscular Falls in the Elderly

### 5.1. Research Methods for Early Warning of Neuromuscular Falls in Elderly

Existing fall-related research generally includes fall prediction and fall detection. The former mainly uses protective devices to reduce damage to the body when a fall occurs, such as some scene recognition systems, visual sensors, acoustic sensors, and wearable sensor equipment. The latter focuses on buying time for patients after the injury occurs, sending out distress signals in time, and obtaining medical assistance. For example, the intelligent system based on floor vibration data proposed by Clemente et al. [67] is for fall detection, location, and notification. The fall event can be recognized immediately by running in real-time, and the sensor is installed on the floor without cameras or other devices that may invade privacy. It can be used in a home environment. However, because the fall has already occurred, fall detection can only reduce but not reverse the severity of injuries. The current research mostly focuses on fall prediction, which can receive signals before the fall event is about to occur, or even a little earlier to help predict and give an early warning of an impending fall [48].

A fall warning system based on sensors, such as sEMG, accelerometers, gyroscopes, air pressure and inertial sensors, stands out among fall prediction systems due to its non-invasive nature, real-time outdoor monitoring capability, low deployment costs, and privacy protection [68,69]. In terms of sensor placement, there are two categories: wearable and non-wearable sensors. The early warning system can not only greatly reduce the occurrence of falls but also prevent the occurrence of fall-related injuries from the root cause. With the continuous development of wearable devices, early warning devices can be connected to various parts of the body, such as the waist, wrist, or ankle, and they can also be used in pendants, watches, mobile phones, and other equipment [70]. The warning system can be very small and light, and the sensor has its own battery power and is waterproof and shockproof [34]. And as the wearable devices continue to evolve, early warning devices can be smaller and lighter and will hardly cause any hindrance when applied to daily activity monitoring.

The fall warning system based on inertial sensors has a high sensitivity. Choi et al. [71] developed a new classification algorithm based on deep learning, using a single inertial measurement unit including an accelerometer and gyroscope sensors, and they involved 34 young volunteers to accurately classify fall, near-falling, and activities of daily living, and the accuracy rate of the improved model reached 99%. However, it takes 400 ms for the inertial sensor to detect a fall event, and the detection time for fall prediction must be less than 200 ms [72], and the inertial sensor is prone to false alarms [73]. Mohawish et al. [74] proposed a wireless fall detection system integrating VSMS and 3D accelerometer sensors positioned on the abdomen. Their prototype achieved a 98.5% accuracy in fall detection through experiments involving 200 samples of daily activities and simulated falls, with real-time alerts transmitted via GSM/GPS modules. Hemmatpour et al. [75] proposed an accelerometer-based human gait analysis mode, an algorithm for real-time fall prediction through gait modeling, which can detect the transition from a normal to an abnormal walking mode with an accuracy rate of 99.2%. However, the fall warning system based on the acceleration sensor can distinguish between static activities and dynamic activities, but it is difficult to distinguish between passive and active movements and it is not suitable for scenarios such as elevators, cars, and subways. Paramasivam et al. [76] combined accelerometer and gyroscope sensors to build a fall warning system. When a fall event is about to occur, an early warning will be issued, and there will be 10 s in the middle to distinguish whether it is a false fall or a message will be sent to the guardian, but gyroscope sensors are susceptible to interference and are expensive.

In addition, there are also studies on the joint use of multiple sensors. For example, it is proposed that the acceleration of the human body exceeding the threshold in a short period of time is considered to be a fall, but because the physiological process is affected by many factors, only observing the acceleration cannot accurately reflect the state of the human body, and the accuracy of using a single indicator is low [77]. Therefore, multiple indicators, such as three-axis acceleration, angle, and gyroscope, are used for human body posture recognition in order to minimize data anomalies caused by factors other than falls to improve the accuracy of the equipment [45]. Another fall detection method for the elderly using nursing aids integrated with multi-array flexible tactile sensors has been developed. The study shows that in safe movement states (e.g., walking, bending over, standing), the recognition accuracy reaches 100%. For fall events, the prediction accuracy is 95% when the lead time exceeds 300 ms, and 97.5% when the lead time is between 200 and 300 ms. After applying information fusion algorithms, the fall prediction accuracy achieves 100%, and with a lead time of over 300 ms, the recognition rate is 96.25%. However, the study notes that the model may have limitations in long-term fall prediction and suggests future work to enhance the model’s effectiveness for elderly ambulation fall detection [78].

In addition to various sensors, sEMG is another important method applied in the fall warning system [68]. The sEMG signal is the biological current generated by the contraction of the surface muscles of the human body, which is the result of the comprehensive superposition of the action potential sequence issued by multiple motor units on the skin surface in time and space, and the sEMG signal is collected by placing the electrode sheet on the skin surface of the corresponding muscle. Proper electrode placement is critical for ensuring a high signal quality and minimizing crosstalk from adjacent muscles. To obtain accurate and reliable signals, electrodes are typically positioned over the muscle belly, where muscle fibers are most concentrated and the signal amplitude is highest; the exact placement depends on the specific muscle being studied. Prior to electrode placement, the skin should be cleaned and abraded to reduce impedance and enhance the signal quality. After acquisition, the raw sEMG signals undergo a series of processing steps, including amplification, filtering, and analog-to-digital conversion, to remove noise and improve the signal fidelity. The processed sEMG signals can then be analyzed to assess muscle fatigue, evaluate neuromuscular function, and monitor muscle activity during various tasks.

In recent years, research hotspots have also focused on smartphone-based fall recognition systems. Embedded motion sensors based in smartphones, such as accelerometers, gyroscopes, and compasses, provide new methods for fall detection and prevention. Previously, falls were detected based on thresholds, which required a trade-off between false negatives and false positives, and the fall risk thresholds of various studies were not uniform. Hassan et al. [79] proposed a mobile-supported fall prediction framework that is based on the mobile phone acceleration sensor to obtain real-time data. The online fall prediction system running in the background can analyze and process the data, and send a distress signal to family members (indoors) or a medical agency (outdoors). The research shows that the age of the smartphone, the type of the phone, and the sensor model will affect the accuracy of the fall warning system [80]. In addition, because the program runs continuously throughout the day, it is necessary to pay attention to the effective use of battery power. The mobile phone sensors (such as the accelerometer) will be limited by the location of the mobile phone, and it is difficult to recognize when the angle of the mobile phone changes slightly due to the location of the fall; these problems are still to be solved. In addition, in recent years, some methods based on artificial intelligence have been continuously applied in the research of fall warning systems to improve accuracy and efficiency. Perhaps in the future we can develop fall warnings suitable for specific individuals by using artificial intelligence. The system, on the basis of protecting user privacy, can identify impending falls early based on individual characteristics, such as weight, age, height, disease, and recent activities [81]. And the analysis of the physiological characteristics of the elderly shows that the use of intelligent prediction tools can determine the key factors that lead to falls and prevent falls to the greatest extent [82]. More importantly, in addition to the detection itself, the fall warning system is equally important to avoid false alarms. Because false alarms will desensitize users to the alarm, it is necessary to count the number of false alarms to improve this problem to the greatest extent. Fanca et al. [80] minimized false detections in fall prediction systems by applying multiple machine learning algorithms, testing and selecting the most suitable one and ensuring proper data handling. Additionally, using large datasets and optimizing algorithms for real-life scenarios further reduces error rates. Machine learning mainly includes four categories: reinforcement learning, supervised learning, semi-supervised learning, and unsupervised learning. Each category contains multiple machine learning algorithms, and each machine learning algorithm used to detect accidents has its own advantages and disadvantages. Therefore, it should be configured and tested before being applied to the real scene to train the algorithm with the best performance.

Accelerometers and gyroscopes are effective for real-time fall detection in wearable devices by monitoring dynamic movements. However, these sensors are susceptible to interference and have a relatively high false alarm rate. In contrast, sEMG provides more precise information about muscle function, which is valuable for detecting neuromuscular impairments prior to a fall. This makes sEMG particularly important for early interventions and the long-term monitoring of neuromuscular health. However, sEMG requires an accurate electrode placement and may not be as flexible for continuous wearable monitoring during daily activities. Ideally, integrating sEMG with accelerometers and gyroscopes can combine the advantages of both, offering a more comprehensive and effective fall prevention system. This integrated system can not only monitor dynamic movements in real-time but also enhance the accuracy of fall prediction through muscle information analysis, reducing false alarms and providing a more reliable safety assurance for elderly individuals.

### 5.2. The Application of sEMG in the Prediction of Neuromuscular Falls in the Elderly

There have been many studies on fall events based on sEMG. The fall prediction and early warning system based on sEMG sensors monitors muscle activity through non-invasive electrodes placed on the skin surface, which can distinguish active movement from passive movement, and sEMG signals can be detected in 200 ms or less to predict fall events [69] and are compatible with complex and diverse physical environments and can better ensure real-time monitoring. These reasons make sEMG sensors a possible solution to the limitations of other sensors [68]. Due to the advantages of sEMG sensors, such as a fast response, universality, no medical monitoring and negligible infections, sEMG sensors stand out in fall warning systems with many principles.

In this review, we focus on the following aspects: fall prediction principles, techniques, equipment, research objectives, subjects, methods, and the fall types evaluated in fall prediction. In addition, we also looked at indicators such as the sensitivity (ability to detect falls), specificity (ability to avoid false positives), and accuracy (ability to distinguish falls from activities of daily living) of each fall prediction device.

Xi et al. [68] used three healthy volunteers (two males and one female) to simulate the seven activities of standing to squatting, squatting to standing, standing to sitting, sitting to standing, stair ascending, stair descending, and walking (during walking, a custom device is installed on the ankle to simulate a fall). And they recorded the sEMG signals of the gastrocnemius, rectus femoris, tibialis anterior, and semitendinosus muscles of the left lower limb based on the principle of the sEMG classifier to find the sEMG feature type and classification methods to provide recommendations for ADL monitoring and fall detection. Finally, it was concluded that in their experimental setup, a high accuracy rate for the recognition and a minimal computational time for daily activity monitoring and fall detection can be achieved with a specificity of 98.95% and a sensitivity of 98.70%, while also including false positive cases which could occur during the process from standing to sitting.

A wireless EMG acquisition system to collect sEMG signals in real-time has been developed, and experiments were conducted on 20 undergraduate volunteers (10 males and 10 females) [83]. The volunteers did not engage in strenuous exercise a week before the experiment to avoid muscle fatigue or strain. During the experiment, EMG sensors were installed on the front of the tibia, gastrocnemius, vastus medialis, and biceps femoris to collect volunteers’ daily life activities (walking, sitting, standing, and going up and down stairs) and falling (stumbling and slipping). The specificity of the system reaches 92.67% and the sensitivity is 93.71% for the sEMG signals, which can effectively distinguish the activities of daily living from falling events. Xiao et al. [83] proposed a fall detection method based on sEMG and collected sEMG signals from 20 volunteers during normal daily activities and falls. The specificity was 92.67% and the sensitivity was 93.71%, which could effectively distinguish daily activities from falls in about 200 ms.

Leone et al. [84] proposed a new type of wireless smart sock based on surface electromyographic signals. The device assesses the risk of falling by monitoring the contraction of the lateral gastrocnemius muscle and the tibialis anterior muscle. The specificity of the smart socks was 83.8%, the sensitivity was 86.4%, and the accuracy was 82.3%. However, in order to reduce the transmission delay and achieve effective monitoring, it is necessary to integrate the receiver/microcontroller unit in the wearable system, so comfort is relatively lost. But research shows that when the elderly use wearable devices, they will be affected by its usability and comfort. Therefore, while considering the technical feasibility of the equipment, the physiological and psychological factors of elderly users should also be considered [85]. Rescio et al. [86] applied sEMG-based smart socks to monitor the gastrocnemius–tibialis muscle, which can monitor the normal daily activities of the elderly and can also be used for long-term muscle behavior monitoring, fall event recognition, and activation protection systems. The system employs machine learning, specifically Linear Discriminant Analysis, to accurately classify fall risks based on sEMG signals, overcoming the limitations of threshold-based approaches and providing a timely detection with a low computational cost. In addition, there are some studies that combine sEMG signals with other signals to distinguish normal daily activities from fall events. In the study of Xi et al. [87], the sEMG signal was combined with the plantar pressure signal, and four surface EMG electrodes were used to collect the surface EMG signals of the vastus lateralis, tibialis anterior, semitendinosus, and gastrocnemius muscles (see Figure 2 for details). The pressure sensor was used to obtain plantar pressure data, which can provide detailed information about foot motion [88]. The pressure sensor helped improve the system accuracy rate to 98% [89]. We think the main reason is that the latter adds muscle fatigue information. This may be because muscle fatigue refers to the reduction in the muscle contraction and force ability, when muscle fatigue is not found in time, it often causes injury and brings pain and economic burdens. The EMG signal is an early indicator for monitoring muscle fatigue and is an objective means for analyzing muscle fatigue [90].

## 6. Improvement and Prospect of Artificial Intelligence Combined with sEMG in Prediction of Neuromuscular Falls in Elderly

If predictions and early warnings can be achieved before the occurrence of falls, the incidence of falls in the elderly will be greatly reduced. Based on this assumption, we can make improvements to the following limitations:

### 6.1. Lack of Realistic Data

Most existing research relies on young, healthy volunteers simulating daily activities and falls in controlled experimental environments. The key problems with this approach are twofold. Firstly, these studies may not accurately reflect the actual physical conditions, neuromuscular deficits, and cognitive impairments that elderly individuals face. Secondly, because they rely on simulations, there is a risk of signal distortion, making it challenging to fully capture the complex factors contributing to neuromuscular falls in the elderly. Additionally, many of these studies lack access to real fall data from elderly individuals. To address this issue, future research should aim to collect more authentic data from elderly individuals who have experienced falls in real-world scenarios, allowing for a more comprehensive understanding of the problem.

### 6.2. Lack of Consistency in Prediction Methods

Research on predicting falls in the elderly is crucial for preventive measures. However, the current landscape is marred by a lack of consistency in prediction methods. Various models and risk thresholds are used, making it difficult to compare findings across studies. Moreover, the applicability of these models to the elderly population as a whole is questionable. Researchers should work towards standardizing the methods for fall prediction, establishing universally applicable risk thresholds, and ensuring that predictive models are validated specifically for elderly populations. This would facilitate the implementation of effective fall prevention strategies.

### 6.3. Limitations of Fall Warning Systems

Fall warning systems, including those utilizing sEMG sensors, offer promise in identifying fall risks. However, they are not without limitations. Factors such as the sensor location, body contact, pressure, sweat, muscle fatigue, and environmental noise can introduce errors and affect the accuracy of these systems. Researchers should focus on improving sensor technology and signal processing techniques to reduce these sources of error. Additionally, validating these systems in real-world conditions with diverse elderly populations is essential to ensure their reliability and effectiveness in practical use.

### 6.4. Emphasis on Mitigating Consequences Rather than Preventing Falls

Much of the current research primarily focuses on minimizing harm after falls have occurred, rather than preventing falls in the first place. While wearable anti-fall protection devices have been developed to reduce the adverse consequences of falls, they do not address the root causes. Future research should shift its emphasis towards preventive measures, including exercise programs, neuromuscular interventions, and environmental modifications, that aim to reduce the incidence of falls. These prevention strategies should be practical, user-friendly, and widely accessible to the elderly population.

In summary, addressing the insufficiency of the existing research on neuromuscular falls in the elderly requires a multidisciplinary approach, including more realistic data collection, standardized prediction methods, improved fall warning systems, and a stronger focus on prevention rather than mitigation. These efforts will contribute to a more comprehensive understanding of the problem and ultimately enhance the quality of life for elderly individuals.

### 6.5. Improvements

#### 6.5.1. Data Collection and Research Approach

Collecting sEMG signals from elderly individuals with a history of falls, along with a control group of healthy individuals, will be an excellent starting point. This approach allows for a comparison of muscle activity patterns and characteristics between different groups. The use of AI, particularly machine learning algorithms, can help analyze and identify specific muscle groups and patterns associated with falls. This data-driven approach can provide valuable insights into the neuromuscular factors contributing to falls in the elderly.

#### 6.5.2. Portable sEMG Acquisition and Analysis System

Developing a portable sEMG acquisition and analysis system tailored for early warnings in the elderly community is a groundbreaking idea. Such a system could be user-friendly, non-invasive, and easily wearable by elderly individuals. It has the potential to continuously monitor muscle activity and detect changes in real-time, providing early warnings of potential fall risks. This technology could significantly enhance the safety and independence of elderly individuals, especially those with neuromuscular declines.

#### 6.5.3. Integration with Alarm Devices and Exoskeletons

The integration of the sEMG-based early warning system with alarm devices and artificial exoskeletons is a forward-thinking approach. In the event of a detected fall risk, the system could trigger alarms or external support mechanisms, such as exoskeletons, to assist the elderly person in maintaining balance or preventing the fall. This combination of technologies has the potential to be a comprehensive and proactive solution to reduce the occurrence of falls in the elderly.

#### 6.5.4. Novelty and Potential Impact

The proposed research’s novelty lies in its proactive approach to fall prevention. By predicting falls before they happen and intervening in real-time, the system has the potential to significantly reduce the incidence of falls among the elderly. This approach contrasts with many current strategies that primarily focus on post-fall care and harm reduction. Reducing the occurrence of falls can lead to an improved overall quality of life for elderly individuals and reduce the burden on families and healthcare systems.

#### 6.5.5. Social and Economic Implications

Preventing falls in the elderly has broad social and economic implications. Fewer falls mean reduced hospitalizations, lower healthcare costs, and improved well-being for the elderly population. It can also alleviate the burden on caregivers and reduce the need for long-term care facilities. By addressing this critical issue, this research has the potential to enhance the overall health and independence of elderly individuals and contribute to the sustainability of healthcare systems.

#### 6.5.6. Future Clinical Work in Elderly Populations

Future clinical research should focus on the elderly population in real-world settings. Large-scale sEMG signal collection should be conducted in real-life environments, targeting elderly individuals with a history of falls as well as healthy subjects, to construct comprehensive datasets that accurately reflect the physiological condition and fall risk of older adults, thus overcoming the limitations of previous studies. Additionally, exploring the integration of sEMG sensors with other types of sensors should be prioritized to optimize fall alert systems. Research efforts should shift towards proactive fall prevention, such as incorporating sEMG monitoring into exercise programs for the elderly and community-based care, alongside long-term follow-up studies. These initiatives will help bridge the gap between technological development and clinical application.

## 7. Conclusions

In this review, we synthesized the current applications of sEMG combined with AI for the prediction of neuromuscular falls in the elderly. Unlike previous reviews that primarily concentrated on post-fall detection using inertial sensors, our work emphasizes a novel perspective: the integration of sEMG and AI for the proactive early warning of fall events. This approach aims to detect neuromuscular deterioration and instability before a fall occurs, enabling real-time interventions that may substantially reduce fall incidences.

Despite promising advancements, several limitations persist in this field. These include the reliance on simulated activities in younger cohorts rather than real-world data from elderly populations, the heterogeneity in predictive methodologies and thresholds, and practical challenges related to the design and deployment of wearable monitoring systems. Addressing these issues requires the establishment of standardized data collection protocols; the development of robust, validated predictive algorithms specifically tailored to elderly physiology; and the creation of user-centric wearable technologies that balance technical performance with comfort and usability.

By advancing predictive modeling and wearable system development, future research based on this direction has the potential to improve functional independence among the elderly and reduce the socioeconomic burden associated with fall-related injuries.

## Figures and Tables

**Figure 1 healthcare-13-01204-f001:**
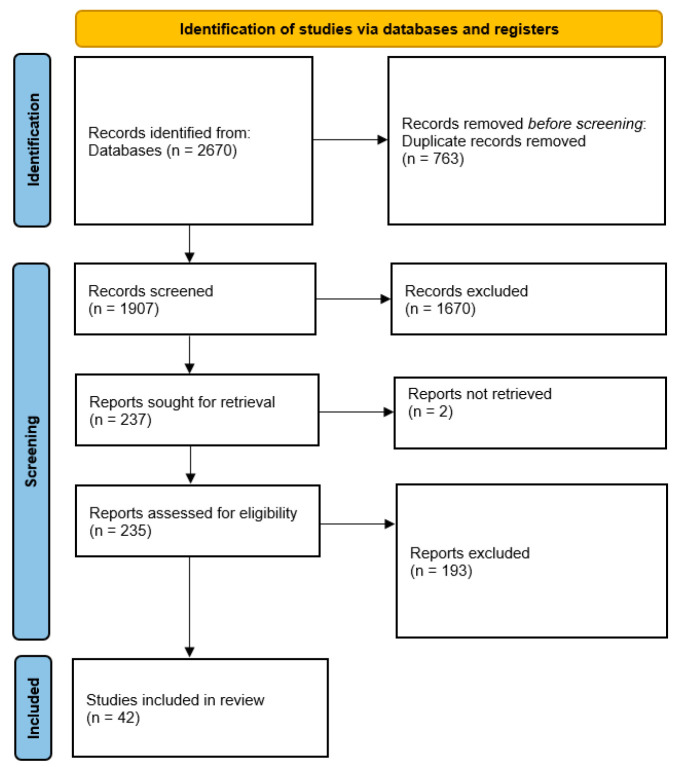
PRISMA flow diagram.

**Figure 2 healthcare-13-01204-f002:**
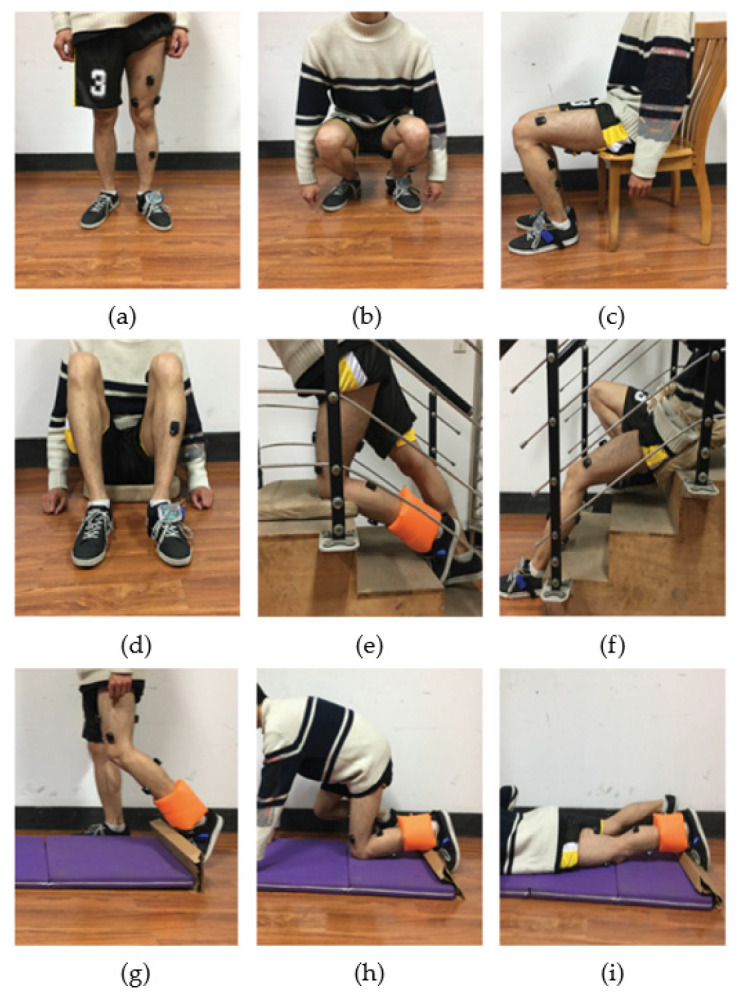
The data collection of the sEMG sensor in combination with the plantar pressure sensor under different postures: (**a**) standing, (**b**) squatting, (**c**) sitting on a chair, (**d**) sitting on the ground, (**e**) going upstairs-falling, (**f**) going downstairs-falling, and (**g**–**i**) walking-falling. Adapted from Xi et al. [87], “Daily Activity Monitoring and Fall Detection Based on Surface Electromyography and Plantar Pressure”, *Complexity*, 2020, CC BY 4.0. Modifications include image rearrangement and labeling simplification. License: [CC BY 4.0] (https://creativecommons.org/licenses/by/4.0/ (accessed on 17 February 2025)).

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
