# Peer review of "Surface Electromyography Combined with Artificial Intelligence in Predicting Neuromuscular Falls in the Elderly: A Narrative Review of Present Applications and Future Perspectives"

_healthcare, 2025, doi:10.3390/healthcare13101204_

Round 1

Reviewer 1 Report

Comments and Suggestions for Authors

This is an interesting review which focuses on the early detection and prevention of falls in the elderly, using electromyography signals to monitor muscle activity. It aims to develop technology that identifies muscle activity abnormalities, enabling timely interventions to enhance muscle strength and stability.

The review is well written, however the authors should address some issues and shortcomings in a major revision, before I could support its publication in this journal.

Abstract:

Reviewer´s comment:

The abstract needs more details about methodology, results and conclusions of this review.

Methods

Reviewer´s comment:

The authors should refer the years in “the range of publication time considered years, from to .”.

When presenting the inclusion and exclusion criteria it should be within the text and not in paragraphs.

The authors should provide the total number of studies included in this review as its area.

Research status of global elderly falls

Reviewer´s comment:

Please correct the citation: “Suey SY Yeung et al. showed that …”.

Please correct the citation: “Research by Bergen, G. et al. showed that in 2014 …”.

Please correct the citation: “According to the research of Moreland, Briana L et al., …”.

Change to 800000: “According to statistics, more than 800,000 patients …”.

Please correct the citation: “Kosse NM et al. found evidence …”.

Application of artificial intelligence and sEMG in the medical field

Reviewer´s comment:

The authors should review the writing. There are some errors in typing such as in punctuation or spelling across the whole chapter.

The authors should correct references to figures.

In “Chen, PJ et al. [53] good designed a wearable sEMG device for lower limb rehabilitation of the elderly at home (as shown in Figure 2).” The authors should correct the citation and change “good designed” to “well-designed”

Insufficiency of existing research on neuromuscular falls in the elderly

The authors should review the writing across the whole chapter.

Please correct the citation: “For example, the intelligent system based on floor vibration data proposed by Clemente J et al …”.

Please correct the citation: “The fall warning system based on inertial sensors has high sensitivity. Choi, A et al. developed …”.

Please correct the citation: “Rihana, S et al. showed …” and add the reference.

Please correct the citation: “Masoud Hemmatpour et al. proposed an accelerometer-based …”.

Please correct the citation: “Paramasivam A et al. combined accelerometer …” and add the reference.

The authors should better explain this paragraph and should provide more detailed information regarding the electrode placing: “The sEMG signal is the biological current generated by the con-traction of the surface muscles of the human body, which is the result of the comprehensive superposition of the action potential sequence issued by multiple motor units on the skin surface in time and space, and the sEMG signal is collected by placing the electrode sheet on the skin surface of the corresponding muscle. After a series of processing, the signals can be used to evaluate the degree of muscle fatigue and muscle function.”

Please correct the citation: “Hassan MM, Gumaei A et al. proposed …”.

Please correct the citation: “Fanca A et al. minimized …”.

Please correct the citation: “Xi X, Tang M et al. [56] used …”.

Please correct the citation: “Leone, Alessandro et al. proposed a new …”.

Please correct the citation: “Rescio G et al. applied sEMG-based …”.

Please correct the citation: “In the study of Xi X, Jiang W et al. [56], the …”.

Comments on the Quality of English Language

The English need to be improved with a english professional editor.

Author Response

Abstract:

Reviewer´s comment:

The abstract needs more details about methodology, results and conclusions of this review.

Response:

We have revised the abstract to include additional details about the methodology, results, and conclusions of this review. 

Methods

Reviewer´s comment:

The authors should refer the years in “the range of publication time considered years, from to .”.

When presenting the inclusion and exclusion criteria it should be within the text and not in paragraphs.

The authors should provide the total number of studies included in this review as its area.

Response:

We have explicitly specified the range of publication years considered in the study within the methods section.

We have revised the presentation of the inclusion and exclusion criteria to ensure they are integrated into the main text rather than being listed in separate paragraphs.

We have added the total number of studies included in this review to clarify the scope of our analysis.

Research status of global elderly falls

Reviewer´s comment:

Please correct the citation: “Suey SY Yeung et al. showed that …”.

Please correct the citation: “Research by Bergen, G. et al. showed that in 2014 …”.

Please correct the citation: “According to the research of Moreland, Briana L et al., …”.

Change to 800000: “According to statistics, more than 800,000 patients …”.

Please correct the citation: “Kosse NM et al. found evidence …”.

Response:

We have revised all the citations and the numerical format you pointed out to ensure proper formatting and clarity.

Application of artificial intelligence and sEMG in the medical field

Reviewer´s comment:

The authors should review the writing. There are some errors in typing such as in punctuation or spelling across the whole chapter.

The authors should correct references to figures.

In “Chen, PJ et al. [53] good designed a wearable sEMG device for lower limb rehabilitation of the elderly at home (as shown in Figure 2).” The authors should correct the citation and change “good designed” to “well-designed”

Response:

We have thoroughly reviewed the entire chapter and corrected all identified errors in punctuation and spelling.

We have reviewed and corrected all references to figures to ensure they are accurate and consistent.

We have corrected the citation in the sentence about Chen, PJ et al. and changed “good designed” to “well-designed” as suggested.

Insufficiency of existing research on neuromuscular falls in the elderly

Reviewer´s comment:

The authors should review the writing across the whole chapter.

Please correct the citation: “For example, the intelligent system based on floor vibration data proposed by Clemente J et al …”.

Please correct the citation: “The fall warning system based on inertial sensors has high sensitivity. Choi, A et al. developed …”.

Please correct the citation: “Rihana, S et al. showed …” and add the reference.

Please correct the citation: “Masoud Hemmatpour et al. proposed an accelerometer-based …”.

Please correct the citation: “Paramasivam A et al. combined accelerometer …” and add the reference.

The authors should better explain this paragraph and should provide more detailed information regarding the electrode placing: “The sEMG signal is the biological current generated by the con-traction of the surface muscles of the human body, which is the result of the comprehensive superposition of the action potential sequence issued by multiple motor units on the skin surface in time and space, and the sEMG signal is collected by placing the electrode sheet on the skin surface of the corresponding muscle. After a series of processing, the signals can be used to evaluate the degree of muscle fatigue and muscle function.”

Please correct the citation: “Hassan MM, Gumaei A et al. proposed …”.

Please correct the citation: “Fanca A et al. minimized …”.

Please correct the citation: “Xi X, Tang M et al. [56] used …”.

Please correct the citation: “Leone, Alessandro et al. proposed a new …”.

Please correct the citation: “Rescio G et al. applied sEMG-based …”.

Please correct the citation: “In the study of Xi X, Jiang W et al. [56], the …”.

Response:

We have corrected all citations you pointed out.

We have revised the paragraph about the sEMG signal to provide more detailed information regarding electrode placement and signal processing.

Reviewer 2 Report

Comments and Suggestions for Authors

This paper presented a review for the use of the sEMG signs in the fall prediction for elderly people with the systems of artificial intelligence. The gap of this study is not really defined. For example, in which review differs from the others reviews or surveys? What are the contributions of this literature review instead of the others?

  • Authors must follow the journal’s template as well as the English. For example, missing punctuation and spelling.
    • “the global population aged 60 and above is expected to reach 2 billion According to WHO”.
    • “portable, Wearable devices serve the elderly population”
    • “Suey SY Yeung et al. showed”
  • Abstract provides information about the background but there is missing information about the methodology of the review, how the search of the papers was conducted, which were the main results and analysis of the authors.
  • The relation between osteoporosis and sEMG is not too related as could be used as motivation. It appears that be used not properly. Authors should analyze the real motivation of using the sEMG for elderly. For example, fibromyalgia and other diseases.
  • The definition of sEMG is not corrected. It should be revised. Moreover, it was explained more than once in the text and in different sections and ways.
  • Several methodological errors in the review. For example, a method was not defined, the years of the review are not presented, the authors did not explain if the terms were combined or not, which procedure was performed for duplicate papers, how much papers were found.
  • A quantitative analysis is not performed by the authors. It is only descriptive review of the literature but not a critical revision or survey, which could indicate trends for the readers.
  • Figures are not mentioned properly in the text. For example, figure 1 was not mentioned. The figures also did not have their sources referenced.
  • The discussion ended abruptly with the paper. It occurred because the revision was not analytical, only a literature review.

I considered that the paper should not be accepted. The authors must provide a contribution in the paper, not only a description of the papers. A quantitative analysis must be performed.

Comments on the Quality of English Language
  • Authors must follow the journal’s template as well as the English. For example, missing punctuation and spelling.
    • “the global population aged 60 and above is expected to reach 2 billion According to WHO”.
    • “portable, Wearable devices serve the elderly population”
    • “Suey SY Yeung et al. showed”

Author Response

Comments 1: This paper presented a review for the use of the sEMG signs in the fall prediction for elderly people with the systems of artificial intelligence. The gap of this study is not really defined. For example, in which review differs from the others reviews or surveys? What are the contributions of this literature review instead of the others?

Response 1: Thank you for your valuable comment. We have clarified the specific novelty and contribution of our review. Compared to previous surveys, our review uniquely focuses on the predictive value of sEMG combined with AI for early warning of neuromuscular falls in the elderly.

Comments 2: Abstract provides information about the background but there is missing information about the methodology of the review, how the search of the papers was conducted, which were the main results and analysis of the authors.

Response 2: Thank you for your insightful comments regarding the abstract. We have revised the abstract to include more comprehensive information about the methodology, results, and analysis.

Comments 3: The relation between osteoporosis and sEMG is not too related as could be used as motivation. It appears that be used not properly. Authors should analyze the real motivation of using the sEMG for elderly. For example, fibromyalgia and other diseases.

Response 3: Thank you for your insightful comments regarding the motivation for using sEMG in elderly individuals. We have revised the manuscript to better address the relationship between sEMG and common elderly diseases such as sarcopenia and fibromyalgia.

Comments 4: The definition of sEMG is not corrected. It should be revised. Moreover, it was explained more than once in the text and in different sections and ways.

Response 4: Thank you for your comments regarding the definition of sEMG. We have revised the definition to ensure it is accurate and comprehensive. We have also streamlined the presentation of sEMG throughout the text to avoid repetition.

Comments 5: Several methodological errors in the review. For example, a method was not defined, the years of the review are not presented, the authors did not explain if the terms were combined or not, which procedure was performed for duplicate papers, how much papers were found.

Response 5: Thank you for your comments. In response to your comment, we have revised the Methods section by providing a detailed description of the review process.

Comments 6: A quantitative analysis is not performed by the authors. It is only descriptive review of the literature but not a critical revision or survey, which could indicate trends for the readers.

Response 6: Thank you for your comments. We acknowledge that this review is descriptive and not a systematic review. Its purpose is to summarize the current applications of sEMG in fall events among the elderly to provide a basis for future research.

Comments 7: Figures are not mentioned properly in the text. For example, figure 1 was not mentioned. The figures also did not have their sources referenced.

Response 7: Thank you for your comments regarding the figures. We have revised the manuscript to ensure all figures are properly mentioned in the text and have added references to their sources.

Comments 8: The discussion ended abruptly with the paper. It occurred because the revision was not analytical, only a literature review.

Response 8: We acknowledge that the previous version ended too abruptly. To address this, we have added a Conclusion section that synthesizes the key findings, highlights the gaps in current research, and proposes future directions.

Comments 9:Authors must follow the journal’s template as well as the English. For example, missing punctuation and spelling.

“the global population aged 60 and above is expected to reach 2 billion According to WHO”.

“portable, Wearable devices serve the elderly population”

“Suey SY Yeung et al. showed”

Response 9: Thank you for your comments. We have revised the manuscript to correct punctuation, spelling, and formatting issues, and have ensured it adheres to the journal’s template.

Reviewer 3 Report

Comments and Suggestions for Authors

Reviewer Comments:

  1. Abstract
  • The objective of the review should be included
  1. Tittle
  •  add type of study
  1. Introduction
  • Include some risk factors, causes/consequences of falls. Both internal and external factors.
  • Further contextualisation of the use of EMG in this research topic is needed.
  • The use of artificial intelligence combined with EMG needs to be better contextualised.
  1. Methods
  • How were the keywords combined in the search?
  • Were mes and not mesh terms used?
  • This area needs to be improved. Add sections with the design of the study, selection criteria... diagram with the initial, final and selected articles...
  1. Application of artificial intelligence in the medical field
  • To deepen the use of artificial intelligence and its relationship with EMG, so that it has applicability to the research topic (care of the elderly).
  1. Researches of sEMG
  •  To deepen the use of EMG as a predictor, rehabilitation or feedback system in the elderly for the prevention of falls or injuries caused by falls.
  1. Research methods for early warning of neuromuscular falls in the elderly
  • Use of other devices in this research topic, such as accelerometers or gyroscopes. benefits of these assessment systems over EMG or vice versa.
  1. Conclusion
  • Add final conclusions regarding the objective set out in the study

Author Response

Abstract

Comments 1: The objective of the review should be included

Response 1: We have added the objective of the review to the manuscript.

Tittle

Comments 2: add type of study

Response 2: I have added the type of this study and modified the title as “Surface electromyography combined with artificial intelligence in predicting neuromuscular falls in the elderly: A Narrative Review of Present Applications and Future Perspectives”

Introduction

Comments 3: Include some risk factors, causes/consequences of falls. Both internal and external factors.

Response 3: We have added information on fall risk factors, including internal factors and external factors. We also discuss the consequences of falls.

Comments 4: Further contextualisation of the use of EMG in this research topic is needed.

Response 4: We have added contextualization of sEMG in this research topic. sEMG is a non-invasive technique used in clinical medicine, rehabilitation, sports science, and ergonomics for neuromuscular assessment and monitoring. In fall prediction among the elderly, sEMG provides a more objective and dynamic evaluation of neuromuscular function compared to traditional methods.

Comments 5: The use of artificial intelligence combined with EMG needs to be better contextualised.

Response 5: We have added contextualization of the use of AI combined with sEMG in clinical medicine. AI integrated with sEMG has achieved significant advances in neuro-muscular function assessment, motor control analysis, disease diagnosis, and rehabilitation interventions.

Methods

Comments 6: How were the keywords combined in the search?

Response 6: A search string consisting of the following keyword combinations was adopted: “elderly” AND (“artificial intelligence” OR “AI” OR “machine learning” OR “deep learning” OR “neural networks”) AND (“sEMG” OR “surface electromyography”) AND “fall”.

Comments 7: Were mes and not mesh terms used?

Response 7: We did not use MeSH terms in the search strategy. Instead, we employed a combination of relevant keywords to ensure a comprehensive retrieval of studies.

Comments 8: This area needs to be improved. Add sections with the design of the study, selection criteria... diagram with the initial, final and selected articles...

Response 8: We have improved the section by providing a detailed description of the study design, selection criteria, and the process of identifying and selecting studies.

Application of artificial intelligence in the medical field

Comments 9: To deepen the use of artificial intelligence and its relationship with EMG, so that it has applicability to the research topic (care of the elderly).

Response 9: We have expanded the discussion on AI and sEMG integration. AI enhances sEMG analysis by automating signal processing and identifying patterns, enabling precise assessments of neuromuscular activity. This approach supports rehabilitation, personalized interventions, and health monitoring, helping predict risks and improve safety in elderly care.

Researches of sEMG

Comments 10: To deepen the use of EMG as a predictor, rehabilitation or feedback system in the elderly for the prevention of falls or injuries caused by falls.

Response 10: We have added content to this section to further explore the use of EMG as a predictor, rehabilitation tool, and feedback system for fall prevention in the elderly.

Research methods for early warning of neuromuscular falls in the elderly

Comments 11: Use of other devices in this research topic, such as accelerometers or gyroscopes. benefits of these assessment systems over EMG or vice versa.

Response 11: We have added content to the last paragraph of this section to compare EMG with accelerometers and gyroscopes, highlighting the benefits of each system in fall prevention.

Conclusion

Comments 12: Add final conclusions regarding the objective set out in the study

Response 12: We have added a final conclusion section that aligns with the study's objective.

Round 2

Reviewer 3 Report

Comments and Suggestions for Authors

The manuscript has been improved from the first version. Some minor revisions

-The flowchart can be improved in terms of quality and content. The reasons for exlusion for example.

-To suggest concrete lines of work for applied clinical research in real populations.

Author Response

Comments 1: The flowchart can be improved in terms of quality and content. The reasons for exclusion for example.

Response 1: Thank you for the helpful suggestion. We have revised the PRISMA flowchart to improve informational content.

Comments 2: To suggest concrete lines of work for applied clinical research in real populations.

Response 2: Thank you for your valuable suggestion to propose concrete lines of work for applied clinical research in real populations. In response, we have added a new subsection to the “6. Improvement and Prospect” section.